# Room-temperature oxygen vacancy migration induced reversible phase transformation during the anelastic deformation in CuO

Lei Li[1,4], Guoxujia Chen[1,4], He Zheng 📵 [1,2,3✉], Weiwei Meng[1], Shuangfeng Jia 📵 [1], Ligong Zhao[1], Peili Zhao[1], Ying Zhang[1], Shuangshuang Huang[1], Tianlong Huang[1] & Jianbo Wang 📵 [1✉]

From the mechanical perspectives, the influence of point defects is generally considered at high temperature, especially when the creep deformation dominates. Here, we show the stress-induced reversible oxygen vacancy migration in CuO nanowires at room temperature, causing the unanticipated anelastic deformation. The anelastic strain is associated with the nucleation of oxygen-deficient $CuO_x$ phase, which gradually transforms back to CuO after stress releasing, leading to the gradual recovery of the nanowire shape. Detailed analysis reveals an oxygen deficient metastable $CuO_x$ phase that has been overlooked in the literatures. Both theoretical and experimental investigations faithfully predict the oxygen vacancy diffusion pathways in CuO. Our finding facilitates a better understanding of the complicated mechanical behaviors in materials, which could also be relevant across multiple scientific disciplines, such as high-temperature superconductivity and solid-state chemistry in Cu-O compounds, etc.

---

[1] School of Physics and Technology, Center for Electron Microscopy, MOE Key Laboratory of Artificial Micro- and Nano-structures, and Institute for Advanced Studies, Wuhan University, Wuhan, China. [2] Suzhou Institute of Wuhan University, Suzhou, Jiangsu, China. [3] Wuhan University Shenzhen Research Institute, Shenzhen, Guangdong, China. [4]These authors contributed equally: Lei Li, Guoxujia Chen. ✉email: zhenghe@whu.edu.cn; wang@whu.edu.cn

The structural stability of material subjected to intense mechanical duress is the corner stone for ensuring the robust performance in micro-electro-mechanical systems and devices, whereas there is a surging demand for the effective damping system to minimize mechanical vibration or noise[1,2]. Toward this end, the anelasticity, characterized by a delay in the shape recovery after retracting the external stress and thus extensively utilized to dissipate the mechanical energy, has been the focus of research for decades[3–5].

In contrast to conventional anelastic behaviors, which generally involve the cooperative motion of atoms at grain or phase boundaries in polycrystalline or composite materials[6–8], the unexpected remarkable anelasticity was reported in single crystalline (e.g., ZnO) nanowires (NWs) recently[9]. It was speculated that the point defect migration in an inhomogeneous stress field, called as Gorsky effect, might account for the observed damping process in single crystalline materials[10–12]. In nanosized materials, the generalized, nonlinear Gorsky effect associated with short diffusion distance and large stress gradient (under bending), could contribute to a much larger anelastic strain, even at room temperature[9]. Nevertheless, the experimental verification of this mechanism has suffered from the difficulty in directly resolving point defects[13], especially when they are mobile under the external stress. More recently, it was theorized that the localized bond stretching and compression[14] rather than the vacancy migration can induce the anelasticity in single crystalline materials, making the underlying anelastic deformation mechanisms elusive and thereby stunting development for device design and modelling.

In what follows, with the joint efforts of advanced electron microscopy[15–26] and first-principles calculations[27,28], we elucidate the oxygen vacancy migration-mediated anelasticity in single crystalline CuO NWs. Atomistic structural characterization reveals that the bending stress-induced vacancy clustering stimulates the nucleation of metastable CuO$_x$ phase, which gradually transforms back to CuO after removal of the bending load.

Our findings provide the direct atomistic view regarding the point defect migration in nanosized materials at room temperature[9].

## Results and discussion

The bending tests were performed with a Nanofactory EP1000 transmission electron microscopy-scanning tunneling microscopy (TEM-STM) holder inside the TEM[29–31]. After bending stress releasing, the single crystalline CuO NW undergoes an instantaneous elastic strain recovery followed by gradual recovering of anelastic strain (Fig. 1a, b, see also Supplementary Fig. 1 and Supplementary Movie 1). Figure 1c reveals that the large anelastic strains (up to 0.46%) in CuO NWs, comparable to those reported in ZnO NWs (as large as 0.64%), substantially recover in 2–3 min. Moreover, the anelasticity also exists when electron beam (e-beam) is turned off, indicating that it is an intrinsic property of single crystalline CuO NWs (Supplementary Fig. 2). To clarify the controversial anelastic deformation mechanisms in single crystalline materials, further atomic-scale structural characterizations were performed. The high-resolution TEM (HRTEM) image in Fig. 1d (the inserted fast Fourier transformation (FFT) image) shows the formation of a different phase at the compressive end of the CuO NW at the moment when the bending stress is removed (enclosed area in Supplementary Fig. 3). Consistently, the anelastic strain recovery (Fig. 1e) is associated with the gradual transformation of nucleated phase to monoclinic CuO structure (space group: C2/c) (see also Supplementary Fig. 4 and Movie 2). The electron energy-loss spectroscopy (EELS) and energy dispersive spectroscopy (EDS) spectra (red line in Fig. 1f and Supplementary Fig. 5) indicate that the different phase mainly consists of Cu and O elements (designated as CuO$_x$), with distinct electron energy-loss near-edge structures of Cu edge as compared with those of CuO (black line in Fig. 1f) and Cu$_2$O (blue line in Fig. 1f). Additionally, the Cu $L_3/L_2$ ratio of CuO$_x$ (~3.50) lies between those of CuO (~3.54) and Cu$_2$O (~3.48), suggesting that the average copper valence state in CuO$_x$

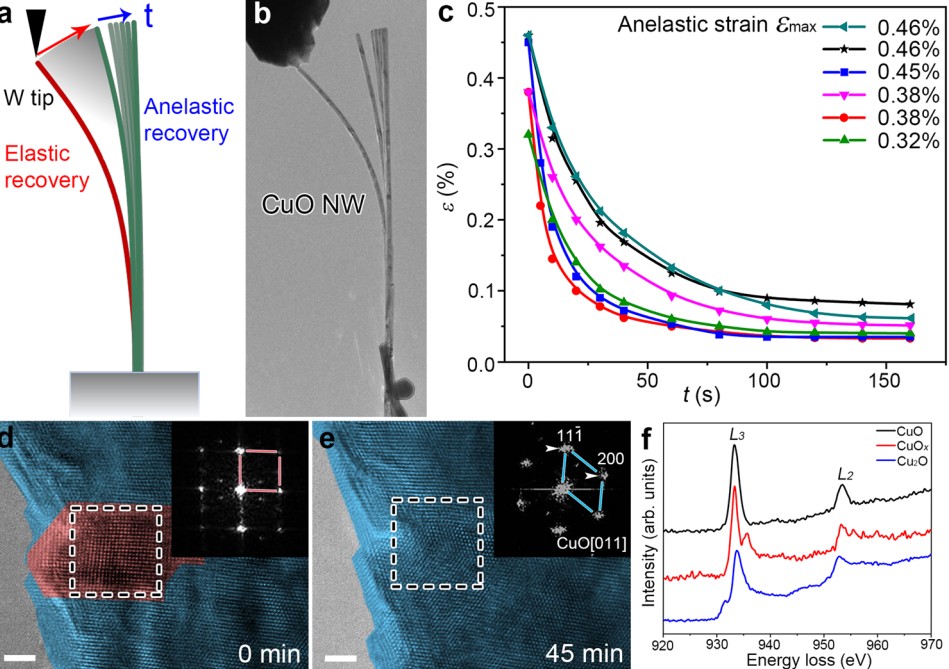

**Fig. 1 The anelasticity in single crystalline CuO NW. a, b** Schematic illustration and experimental observation of the anelastic deformation, respectively. **c** Anelastic strain recovery as a function of time in six bending tests, **d** HRTEM image showing the nucleation of CuO$_x$ phase (red area), which transformed back to CuO (blue area) after 45 min (**e**). Inset: the FFT pattern of enclosed area. **f** The EELS spectra of CuO, CuO$_x$, and Cu$_2$O. Scale bar, 200 nm (**b**), 2 nm (**d, e**).

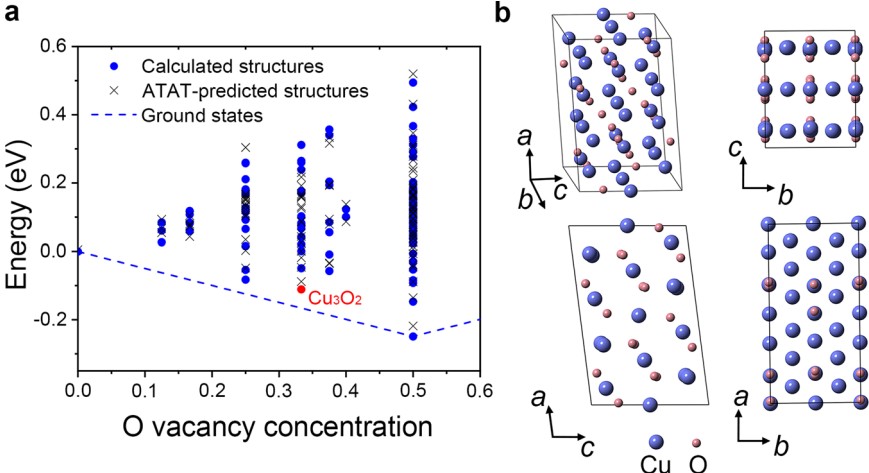

**Fig. 2 The crystal structure of $CuO_{0.67}$ (i.e., $Cu_3O_2$) phase based on first-principles calculations. a** Calculated energies of different $CuO_x$ ($0.5 < x < 1$) phases versus the concentration of O vacancy ranging from 0 to 50%. The dashed lines delineate the theoretical ground states. The blue points and black crosses represent the calculated and ATAT-predicted structures, respectively. The red point refers to $Cu_3O_2$ structure with the lowest energy when the O vacancy concentration is ~33%. **b** The structural models of $Cu_3O_2$ viewed along the $a$, $b$, and $c$ axes.

is between 1+ and 2+, i.e., $x$ ranges from 0.5 to 1[32]. However, to our best, the $CuO_x$ phase (Fig. 1d) cannot be indexed based on any reported Cu–O compound structure reported hitherto.

First-principles calculations combined with cluster expansion construction method[28] were applied to reveal the atomic structure of $CuO_x$. Alloy Theoretic Automated Toolkit (ATAT) is a collection of alloy theory tools[33], which could automate alloy phase diagram calculations by atom substitutions (Fig. 2a). Among all the calculated phases, a typical $CuO_{0.67}$ (i.e., $Cu_3O_2$) phase with unreported monoclinic structure (space group: $C2/m$, $a = 11.49$ Å, $b = 5.75$ Å, $c = 7.49$ Å, and $\beta = 97.18°$, the detailed atom positions are presented in Supplementary Table 1) can be applied to satisfactorily characterize the $CuO_x$ phase (Fig. 2b). To be specific, Supplementary Fig. 6 shows the high angle annular dark field (HAADF) and annular bright field (ABF) images of CuO along the same observation direction in Fig. 1d, i.e., $[011]_{CuO}$. Meanwhile, the nucleated phase can be characterized by the calculated $Cu_3O_2$ structure along $[\overline{1}34]_{Cu_3O_2}$ direction (Supplementary Fig. 6d). The crystallographic orientation relationship (OR) between CuO and $Cu_3O_2$ is

$$\begin{pmatrix} a \\ b \\ c \end{pmatrix}_{Cu_3O_2} = \begin{pmatrix} -1.2831 & 1.3530 & -1.8883 \\ -0.7463 & -1.3447 & -0.1327 \\ -0.8805 & 0.8542 & 0.9528 \end{pmatrix} \cdot \begin{pmatrix} a \\ b \\ c \end{pmatrix}_{CuO} \quad (1)$$

Such OR is maintained when observed along different zone axes including $[1\overline{1}2]_{CuO}$, $[100]_{CuO}$, and $[001]_{CuO}$ axes (Supplementary Figs. 7–9), verifying that the nucleated phase possibly possesses an oxygen-deficient $Cu_3O_2$ structure (compared to CuO). However, further experimental studies are of necessity to confirm the accurate value of $x$. This metastable $CuO_x$ phase is a distorted structure of CuO with O vacancies or $Cu_2O$ with O interstitials. Given that $CuO_x$ phase nucleates under strain and will slowly transform into CuO after the stress released, the local lattice distortion may exist which allows for ~5% error in correlating the experimental and calculated results. It is worth mention that $Cu_3O_2$ phase has been frequently reported during the oxidation of Cu and $Cu_2O$[34,35]. Unfortunately, since $Cu_3O_2$ is unstable and can be easily oxidized to CuO, its structural details remain mysterious due to the lack of atomic-scale structural investigation. The calculated X-ray diffraction (XRD) patterns

based on current $Cu_3O_2$ structural model and the experimental XRD pattern of pulsed laser deposited cuprous oxide thin films[36] are displayed in Supplementary Fig. 10. We found that although there might be some similarity between the two structures, the peaks do not match quite well between the two phases, possibly due to the strong texture in the experimental deposited film. Further efforts are required to reveal the structural relationship between current metastable $CuO_x$ phase and $Cu_3O_2$ phase reported in the literatures[34–36].

The nucleation of oxygen-deficient $CuO_x$ phase in CuO NW could be attributed to the Gorsky effect[10–12], whereas the oxygen vacancies can diffuse from the tensile to compressive side under the inhomogeneous stress field induced by mechanical bending (Supplementary Fig. 3)[9]. Afterwards, removing the bending strain eliminates the stress gradient along the NW radial direction, resulting in the reversed movement of the oxygen vacancies and thus the transformation of $CuO_x$ phase back to CuO (Fig. 1d, e). The experimental results are repeatable when observed along $[101]_{CuO}$ (Supplementary Fig. 11) and $[1\overline{1}2]_{CuO}$ (Supplementary Fig. 12) axes.

More importantly, the possible vacancy diffusion pathway is discussed. The time-lapsed images in Fig. 3a–c illustrate the evolution of phase boundaries between $CuO_x$ phase and CuO along $[011]_{CuO}$ axis. It is evident that the boundaries mainly consist of $(11\overline{1})_{CuO}$, $(1\overline{1}1)_{CuO}$, $(200)_{CuO}$, and $(3\overline{1}1)_{CuO}$ planes (Fig. 3, Supplementary Fig. 4 and Supplementary Movie 3). The diffusion barriers for O vacancies in CuO were calculated through climbing image nudged elastic band (CI-NEB) method[37]. Three possible diffusion pathways with lowest barriers of 1.11, 1.11, and 1.89 eV are along $[1\overline{1}0]_{CuO}$, $[110]_{CuO}$, and $[010]_{CuO}$ directions, respectively (Fig. 3d), larger than the diffusion barrier of O vacancies near the surface along $[110]_{CuO}$ direction (less than 1 eV) reported previously[37]. Accordingly, the structural model of CuO is presented in Fig. 3e, f to better illustrate the diffusion pathways. It is found that the diffusion directions of $[1\overline{1}0]_{CuO}$, $[110]_{CuO}$, and $[010]_{CuO}$ are located in the $(11\overline{1})_{CuO}$, $(1\overline{1}1)_{CuO}$, and $(200)_{CuO}$ planes, respectively, suggesting that the appearance of the three planes as the dominant phase boundary can be well anticipated. Additionally, the $(3\overline{1}1)_{CuO}$ boundary can be considered as the combination of both $(11\overline{1})_{CuO}$ and $(200)_{CuO}$ planes (Supplementary Fig. 13). In addition, according to the previous references[9], the surface-mediated point defect could also occur under the mechanical stress. The CuO NW geometry is further investigated.

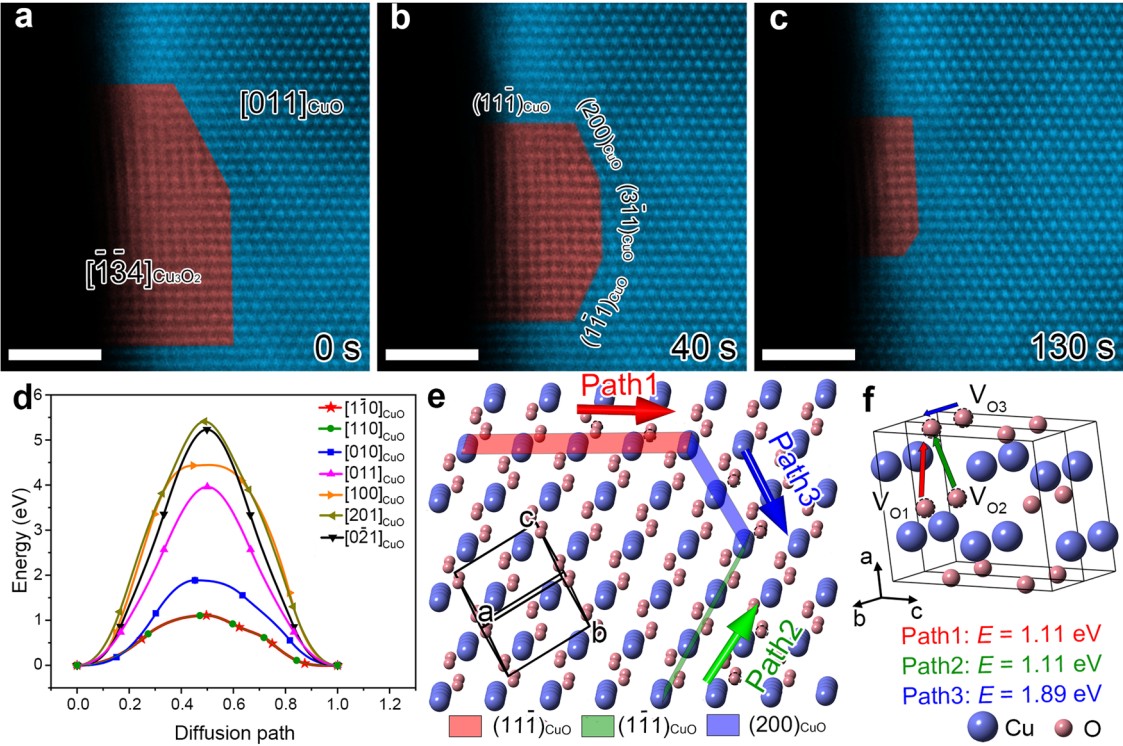

**Fig. 3 The oxygen vacancy diffusion pathways in CuO. a–c** The snapshots showing the phase transition from metastable CuO$_x$ phase to CuO induced by the vacancy migration. **d** CI-NEB calculations of diffusion barriers for O vacancies along different paths in CuO. **e** Three possible diffusion paths of O vacancies in CuO. **f** CuO unit cell showing the possible migration directions of O vacancies with the diffusion barriers listed below. Scale bar, 2 nm (**a–c**).

We found that the NW cross-sections are nonsymmetric and not circular and surfaces are faceted (Supplementary Fig. 14), which are consisted of various lattice planes, e.g., (002), (110), (11$\bar{1}$), and (1$\bar{1}$1) etc. Besides, the (20$\bar{2}$) plane was reported to be another possible surface plane[38]. Hence, we have calculated the diffusion energy barriers of oxygen vacancies along different surfaces. The relatively lower calculated diffusion energy barriers of oxygen vacancies on different surface planes (e.g., (002), (110), (11$\bar{1}$), (1$\bar{1}$1), (20$\bar{2}$), (200), (011), and (1$\bar{1}$2)) of CuO NW (less than 1 eV, Supplementary Fig. 15) than those in the bulk (the lowest migration energy in Fig. 3 is ~1.11 eV) suggests that the oxygen vacancies prefer to migrate along the surface area, which may result in the relatively larger anelasticity in the nanosized material with higher aspect ratio.

It is generally believed that the room-temperature point defect diffusion in solids is limited, yielding a negligible effect on the mechanical behaviors. By contrast, the present experiments provide the direct evidence that the vacancy diffusion could indeed operate in nanomaterials, which could possibly result from enhanced surface-mediated diffusion and reduced vacancy migration energy under ultrahigh stress[9]. As shown in Supplementary Fig. 16, it is evident that the nonlinear Gorsky effect dominates during the anelastic strain recovery, identical to that reported in ZnO NWs[9], suggesting that there are at least two kinds of point defects. Besides oxygen vacancy, Cu vacancy has been reported as another dominant type of point defect in CuO[39]. Considering the migration of both O and Cu vacancies, we obtained the average diffusivities of $D_1 = 4.09 \times 10^{-14}$ cm$^2$/s and $D_2 = 3.39 \times 10^{-15}$ cm$^2$/s for two types of point defects, respectively, close to the values reported for O vacancies and Zn interstitials[9]. Because the experimental NWs are not central-symmetric (Supplementary Fig. 14), and the stress/strain is not evenly distributed along the NW length direction, the calculated values may not be accurate. Besides, the nucleation of

oxygen-deficient CuO$_x$ phase implies that the O vacancy migration may play a major part. The averaged bending stress before the W tip removed is ~1.29 GPa (Supplementary Fig. 17). Based on the previous investigation[7], the e-beam induced temperature rise was estimated to be only 0.4 K. As compared with the high T$_m$ (~1600 K) of CuO, the thermal effect is negligible in mediating the anelasticity in single crystalline CuO NWs. Furthermore, the e-beam irradiation is shown to expedite the shape recovery when repeatedly compressing the same NW (Supplementary Fig. 2). The momentum/energy transferred from the high-energy e-beam (~200 keV) to O atoms could facilitate the oxygen vacancy migration and thus accelerate the anelastic strain recovery[40].

Similar anelastic deformation behavior was detected in twinned CuO NWs (Supplementary Figs. 18–19)[7]. The HRTEM images and inserted FFT patterns (Supplementary Figs. 18d-f) demonstrate the formation of CuO$_x$ phase, which gradually transformed back to CuO after stress release, implying that the point defect can get through a grain or interphase boundary[2], similar to the intercrystalline Gorsky effect[41].

The anelasticity induced by reversible phase transition was also found in Ni-Ti alloys[42], whereas the lattice mismatch between two phases can contribute to the anelastic strain. Here, the shear strain induced by the nucleation of CuO$_x$ phase (central part in Fig. 4a) in original CuO NW is directly visualized and analyzed in the nanobridges fabricated by the top-down method[43]. Obviously, the shear strain slowly decreases with the gradual phase transition from CuO$_x$ phase to CuO (Fig. 4a-c). To describe the process quantitatively, the shear strain maps $\varepsilon_{xy}$ of the nanobridge are presented in Fig. 3d–f. The geometrical phase analysis (GPA) method is listed in Supplementary Fig. 20. The shear strain as large as 12% gradually decreases to 0 after the phase transition, accompanied with the nucleation and escape of dislocations (Supplementary Fig. 21 and Supplementary Movie 4). The

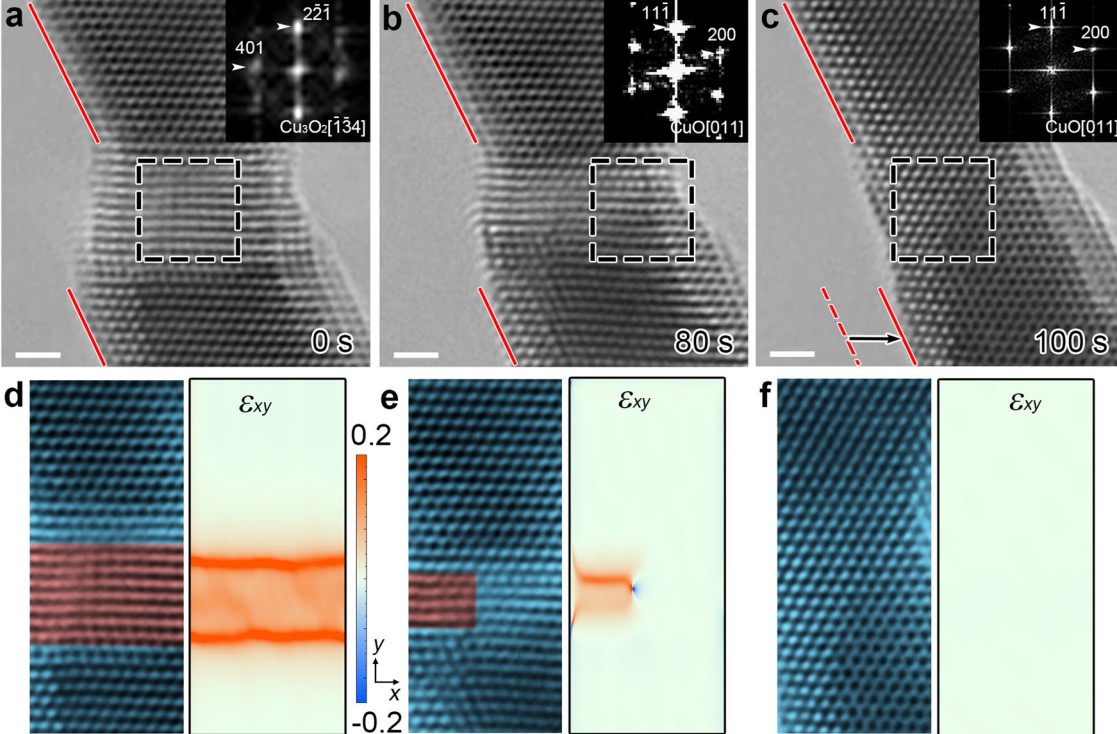

**Fig. 4 The shear strain caused by the phase transition from CuOx phase to CuO. a–c** Sequential HRTEM images showing the transition from CuOx phase to CuO along the $[011]_{CuO}$ axis. **d–f** The corresponding shear strain map ($\varepsilon_{xy}$) of the HRTEM images in **a–c**. Scale bar, 1 nm (**a–c**).

recovery process occurs along the $(11\bar{1})_{CuO}$ boundary (Supplementary Fig. 21a–c), consistent with the oxygen vacancy diffusion pathway calculated in Fig. 3.

The stress-mediated phase transition implies that during the bending process, the chemical bonds between Cu and O (Cu–O) are altered and the structure of CuO is distorted. The slight lattice distortion might affect the magnetic coupling between $3d$ $Cu^{2+}$ spins in Cu–O layers of superconductor Yttrium Barium Copper Oxide (YBCO)[44,45], which plays a pivotal role in understanding the high-temperature superconductivity. For instance, the diffusion of O vacancies in Cu–O layers may also have impact on the superconductivity of YBCO[46,47]. Interestingly, our theoretical calculations (Supplementary Fig. 22) indicate that $Cu_3O_2$ is a half-metallic antiferromagnet, characterized by the spin-polarized electrons near the Fermi level, which may enable single spin superconductivity with triplet Cooper pairs[48].

In summary, we present the first atomistic view of the phase transition from CuO to an oxygen-deficient metastable CuOx phase, as triggered by the stress-gradient-induced oxygen vacancy diffusion from the tensile to the compressive ends. After the stress releasing, the reversible phase transition from CuOx phase to CuO is directly monitored, leading to the gradual recovery of the anelastic strain. Meanwhile, three possible oxygen diffusion pathways along $[1\bar{1}0]_{CuO}$, $[110]_{CuO}$, and $[010]_{CuO}$ are proposed based on both experimental and theoretical studies. The existence of twin boundary does not shut off the anelastic behavior, implying that the oxygen vacancy could diffuse across the boundary. These findings provide the intuitive insight into the point defect-mediated mechanical behaviors in solid materials, consistent with that reported by Cheng et al.[9]. It is suggested that caution should be taken when interpreting the mechanical parameters of nanoscale metal oxides with high strength and inherent point defects when subjected to inhomogeneous stress field, such as bending and torque, etc.

## Methods

**Sample synthesis**. The CuO NWs were prepared through thermal oxidation by heating copper grids/films at about 450 °C for 1–2 h in a muffle furnace. Nanobridges were fabricated by traditional top-down method utilizing the focused e-beam[49].

**In situ TEM testing**. The bending tests were performed on a Nanofactory EP1000 TEM-STM holder inside the TEM. The CuO NWs were attached to tungsten (W) rod with conductive silver paint, which served as one end of the in situ platform. A wedge-shaped W tip acted as the other end of the platform.

**HRTEM and scanning TEM (STEM)/EELS experiments**. In situ HRTEM observations were performed on JEOL F200 and JEOL JEM 2010FEF operated at 200 kV with e-beam intensity of $10–10^2$ A/cm². HAADF, ABF imaging and EDS, EELS analysis were carried out on JEOL ARM200CF operated at 200 kV, with current density on the scan region of around $10–10^2$ A/cm². The acquisition time for each selected pixel for acquiring EELS spectra is less than 1 sec to minimize the irradiation damage.

**HAADF and ABF simulations**. The HAADF and ABF images were simulated by QSTEM software. Based on current experimental conditions, the acceleration voltage of e-beam is set to be 200 kV, with spherical aberration of 0.001 mm, convergence angle of 15 mrad, and the sample thickness ~5 nm.

**Geometrical phase analysis**. The GPA[50] was conducted to generate strain maps of HRTEM images. For the two-dimensional (2D) lattice shown in an HRTEM image, the tensorial distortion is defined as.

$$e = \begin{pmatrix} e_{xx} & e_{xy} \\ e_{yx} & e_{yy} \end{pmatrix} = \begin{pmatrix} \partial u_x/\partial x & \partial u_x/\partial y \\ \partial u_y/\partial x & \partial u_y/\partial y \end{pmatrix} \quad (2)$$

whereas $\mathbf{u} = (u_x, u_y)$ denotes the displacement field. In addition, the shear strain is usually defined as

$$\varepsilon_{xy} = \varepsilon_{yx} = 1/2(\partial u_x/\partial y + \partial u_y/\partial x) \quad (3)$$

The analysis in this work was done by open-source software Strain + +. The details including nonlinear g vectors used for GPA are presented in Supplementary Fig. 20.

**Simulation details**. First-principles calculations were carried out using the Vienna ab initio simulation package (VASP)[51], with the projector augmented wave

method[52,53] and the Perdew-Burke-Ernzerhof (PBE)[54] exchange-correlation functionals. A 450-eV energy cutoff and a $U_{eff}$ of 7 eV in the PBE + U method were used throughout the calculations. ATAT[33] was used to automate first-principles phase diagram calculations to find possible $CuO_x$ phases. The Monkhorst–Pack scheme of k-point mesh was chosen such that the number of k-points times the number of atoms in the unit cell being 1000, which keeps the density of k-point constant regardless of supercell size.

**Calculation and discussion of the bending strain in CuO NW.** The bending strain $\varepsilon$ in a bent NW can be calculated by the formula

$$\varepsilon = \frac{d}{2\rho} = \frac{d\kappa}{2} \quad (4)$$

where $d$ is the diameter of the curved NW, $\rho$ is the NW's curvature radius, and $\kappa$ is the curvature. To calculate the strain in the NW, the NW's shape should be fitted into a mathematical curve by polynomial fitting with the curvilinear equation $y = f(x)$. Then the curvature can be calculated by the formula.

$$\kappa = \frac{|y''|}{(1 + y'^2)^{3/2}} \quad (5)$$

where $y' = f'(x)$ and $y'' = f''(x)$, which represent the first and second derivative of $y = f(x)$, respectively.

**Calculation of diffusivities of point defects in CuO NW.** According to the Gorsky effect or nonlinear Gorsky effect, the diffusivity of point defects could be estimated by measuring the relationship between strain and time as well as considering the geometry of NWs. The relationship of the anelastic strain and time is approximately described by an exponential law in the Gorsky relaxation process. The logarithm transform of anelastic strain would be related to time linearly, with the slope being $1/\tau$ (the relaxation time $\tau$). For cylindrical specimens with the diameter of $d$, the diffusivity $D$ is related to the relaxation time $\tau$ by[12]

$$13.55D \cdot \tau = d^2 \quad (6)$$

Take the NW shown in Fig. 1b for an example ($d \sim 46$ nm), we obtain the average diffusivities of $D_1 = 4.09 \times 10^{-14}$ cm$^2$/s and $D_2 = 3.39 \times 10^{-15}$ cm$^2$/s.

## Data availability

The authors declare that the data supporting the findings of this study are available within the article and the corresponding Supporting Information file. All other relevant source data are available from the corresponding author upon reasonable request.

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

## Acknowledgements

This work was supported by the National Natural Science Foundation of China (52071237, 12074290, 51871169, 51671148, 11674251, and 51601132), the Natural Science Foundation of Jiangsu Province (BK20191187), the Fundamental Research Funds for the Central Universities (2042019kf0190), the Science and Technology Program of Shenzhen (JCYJ20190808150407522), and the China Postdoctoral Science Foundation (2019M652685).

## Author contributions

H.Z. and J.W. conceived and designed the project, L.L., L.Z., P.Z., Y.Z., T.H., S.H., and S.J. carried out the experiments. G.C. and W.M. performed the simulations. L.L., G.C., H.Z., and J.W. analyzed data and wrote the manuscript. All the authors participated in the discussions of the research.

## Competing interests

The authors declare no competing interests.
