## [Peer Review File · Nature Communications]

REVIEWER COMMENTS

Reviewer #1 (Remarks to the Author):

The authors reported a reversible phase transformation in CuO nanowires during bending, which was attributed to migration of oxygen vacancies as a result of heterogeneous strain distribution. A new oxygen-deficient Cu oxide was identified based on atomic resolution imaging and EELS analysis. The diffusion pathways of oxygen vacancies in CuO were proposed based on atomistic calculation and experimental observations. Overall, it is an interesting finding to confirm the migration of point defects at room temperature, which leads to the anelastic behavior of one-dimensional nanomaterials, especially nanowires. The manuscript is suggested to be accepted after taking into account of the following comments or suggestions.

>> The authors wrote in the Introduction “Nonetheless, such deformation mechanism remains controversial since it contradicts with the well-documented principle that the influence of point defects on mechanical properties should be negligible at a temperature much lower than the material melting point (T_m).” This statement is confusing and potentially misleading. On one hand, the present work discusses the oxygen vacancy migration-mediated anelasticity, following essentially the same framework in refs 9-12 about point defect diffusion. On the other hand, the statement that “the influence of point defects on mechanical properties should be negligible at a temperature much lower than the material melting point (T_m)” has its own limitations. The statement holds true at the bulk scale (large diffusion distance) under low stress and/or stress-gradient level. That’s why the anelasticity induced by point defect diffusion is negligible (but measurable), as reported in refs 10-12. But it does not mean the Gorsky effect is controversial. Rather the Gorsky effect is universally true as a result of diffusion, as long as there are stress gradient and diffusion species. Ref 9 significantly expands the scope of the Gorsky effect. In nanostructures, due to short diffusion distance and large stress-gradient (under bending), the anelasticity can become much larger than the classic Gorsky effect, even at room temperature. Ref 9 also firmly establishes the generalized, nonlinear Gorsky theory, derived from the diffusion equation (the classic Gorsky theory is a linear approximation in the case of small stress gradient). Therefore, the point defect diffusion-based deformation mechanism (either the classic, linear or the generalized, nonlinear Gorsky effect) is not controversial. To my understanding, the present work indeed confirms the point defect diffusion at room temperature and additionally reports a plausible pathway following the point defect diffusion – phase transformation – in the case of CuO nanowires. The authors should rewrite their Abstract, Introduction and possibly Discussion to reflect the point raised here.

>> It has been very challenging to measure diffusivity of point defects at room temperature. In this work, with a sequence of TEM images, it might be possible to measure diffusivity. This could be a missed opportunity that the authors can look into.

>> The nanowire in Figure 4 appears to be much smaller. Is it the same nanowire as the one in Figure 1?

>> The authors demonstrated that the new phase CuO_x was not stable. However, details were shown in

Figures 3 and 4. It should be clarified that how the authors kept such phase in the nanowire for further study like EELS and EDS analysis. It should be mentioned (or more details should be given) if the experimental conditions were changed for different analysis.

>> Surface-mediated migration of point defects could be dominated in nanowires during anelastic deformation. A remaining question that was not clearly addressed in this work is - does such migration of point defects promote the formation of the metastable phase CuOx in the present study? For example, Figure 4 clearly verified the formation of the metastable phase CuOx but it is unclear if the CuOx is related to the anelastic deformation.

>> The authors gave an example showing the new phase formed very close to one end of the nanowire. For a bending test, the strain (or stress) distribution should be uniformly distributed along the nanowire length direction. Is there any evidence showing that the new phase is distributed along the nanowire length direction?

>> For the identification of new phase CuOx, there is no direct experimental evidence to support that x is 0.67. It is suggested to use "metastable phase" to describe the experimental findings, although the stoichiometry of the new phase could be Cu3O2.

>> Page 11, line 208, "oxygen diffusion pathways ... are revealed", where more accurately "revealed" is suggested changing to "proposed".

Reviewer #2 (Remarks to the Author):

In my opinion this is an interesting and original paper, worthy of publication. It is generally well written with only a few minor changes needed to the style of the English used.

The identification of the new Cu3O2 phase appears to be reasonably convincing, but it is the manner by which it is formed and the reversal back to CuO, that I have some issues with. In particular, the authors state that the nucleation of the new phase may be due to the Gorsky effect in which oxygen vacancies diffuse from the tensile to the compressive region of the sample during mechanical loading. They also claim that this movement is reversed after the load is removed. However, they appear to be neglecting the geometry of the nano-wire, with many free surfaces in the vicinity of the transformed region. These surfaces can act as sources and sinks for vacancies; hence on removal of the load, in particular, oxygen vacancies could readily move to these nearby surfaces, even at room temperature. Hence, I believe that the paper needs to be modified to take greater account of the role of the sample geometry in any discussion of the mechanisms.

Some comparison of this newly found Cu3O2 phase with the phases formed during the deposition of PLD films of copper oxide is also made in the paper. (Supplementary Fig 10 and reference 1). I find this comparison unconvincing, since only one XRD peak shown in Fig. 10 is common in the two structures.

Other peaks are found in one and not the other and vice versa. This could be explained, in part, by a strong texture in the deposited film, but although there may be some commonality between the two observations, I believe that the comparison is overstated in the paper.

Response to the reviewers

Reviewer #1:

The authors reported a reversible phase transformation in CuO nanowires during bending, which was attributed to migration of oxygen vacancies as a result of heterogeneous strain distribution. A new oxygen-deficient Cu oxide was identified based on atomic resolution imaging and EELS analysis. The diffusion pathways of oxygen vacancies in CuO were proposed based on atomistic calculation and experimental observations. Overall, it is an interesting finding to confirm the migration of point defects at room temperature, which leads to the anelastic behavior of one-dimensional nanomaterials, especially nanowires. The manuscript is suggested to be accepted after taking into account of the following comments or suggestions.

1. The authors wrote in the Introduction “Nonetheless, such deformation mechanism remains controversial since it contradicts with the well-documented principle that the influence of point defects on mechanical properties should be negligible at a temperature much lower than the material melting point (T_m).” This statement is confusing and potentially misleading. On one hand, the present work discusses the oxygen vacancy migration-mediated anelasticity, following essentially the same framework in refs 9-12 about point defect diffusion. On the other hand, the statement that “the influence of point defects on mechanical properties should be negligible at a temperature much lower than the material melting point (T_m)” has its own limitations. The statement holds true at the bulk scale (large diffusion distance) under low stress and/or stress-gradient level. That’s why the anelasticity induced by point defect diffusion is negligible (but measurable), as reported in refs 10-12. But it does not mean the Gorsky effect is controversial. Rather the Gorsky effect is universally true as a result of diffusion, as long as there are stress gradient and diffusion species. Ref 9 significantly expands the scope of the Gorsky effect. In nanostructures, due to short diffusion distance and large stress-gradient (under bending), the anelasticity can become much larger than the classic Gorsky effect, even at room temperature. Ref 9 also firmly establishes the generalized, nonlinear Gorsky theory, derived from the

diffusion equation (the classic Gorsky theory is a linear approximation in the case of small stress gradient). Therefore, the point defect diffusion-based deformation mechanism (either the classic, linear or the generalized, nonlinear Gorsky effect) is not controversial. To my understanding, the present work indeed confirms the point defect diffusion at room temperature and additionally reports a plausible pathway following the point defect diffusion – phase transformation – in the case of CuO nanowires. The authors should rewrite their Abstract, Introduction and possibly Discussion to reflect the point raised here.

Reply: We thank the reviewer for the insightful comments and suggestions. We notice that our previous statement could be misleading. Accordingly, we have rewritten the related contents in Abstract, Introduction and Discussion parts. As suggested by the reviewer, we focus on the discussion that “the present work indeed confirms the point defect diffusion at room temperature”, which provides a plausible way to investigate the point defect diffusion mechanisms.

(1) Replacing “Our finding constitutes a missing piece of the puzzle for interpreting the complicated mechanical behaviors in materials,” by “Our finding facilitates a better understanding of the complicated mechanical behaviors in materials,” (*see line 10, paragraph 1, page 1 in the revised manuscript*)

(2) Replacing “Nonetheless, such deformation mechanism remains controversial since it contradicts with the well-documented principle that the influence of point defects on mechanical properties should be negligible at a temperature much lower than the material melting point (T_m).” by “In nano-sized materials, the generalized, nonlinear Gorsky effect associated with short diffusion distance and large stress-gradient (under bending), could contribute to a much larger anelastic strain, even at room temperature.” (*see line 7, paragraph 2, page 2 in the revised manuscript*)

(3) Replacing “Our finding highlights the importance of the point defect in mediating the mechanical behaviors at room temperature.” by “Our findings provide the direct atomistic view regarding the point defect migration in nano-sized materials at room temperature⁹.” (*see line 6, paragraph 2, page 3 in the revised manuscript*)

(4) Replacing “These findings provide the intuitive insight into the point defect-mediated mechanical behaviors in solid materials.” by “These findings provide the intuitive insight into the point defect-mediated mechanical behaviors in solid materials, consistent with that reported by Cheng *et al*.” (see line 9, paragraph 1, page 13 in the revised manuscript)

2. It has been very challenging to measure diffusivity of point defects at room temperature. In this work, with a sequence of TEM images, it might be possible to measure diffusivity. This could be a missed opportunity that the authors can look into.

Reply: We thank the reviewer for the insightful suggestion. The vacancy diffusivity cannot be directly estimated since the concentration of O vacancies cannot be resolved based on the TEM images, which provide the two-dimensional projection view of a three-dimensional object. However, according to the Gorsky effect or nonlinear Gorsky effect (Gorski, W. S. *Phys. Z. Sowj.* **8**, 457-471 (1935); Cheng, G. *et al. Nat. Nanotechnol.* **10**, 687-691 (2015)), the diffusivity of point defects could be estimated by measuring the relationship between strain and time as well as considering the geometry of nanowires (NWs). The cross-sections of most NWs are not central-symmetric (Figs. R1-R2) which may make the calculated diffusivity through Gorsky effect less credible.

Figure R1 (Supplementary Figure 14) | The morphologies of CuO NWs. a-c SEM images showing the morphologies of fabricated CuO NWs. d-f, TEM images showing the cross-sections of the CuO NWs fabricated by focused ion beam (FIB). Scale bar, 2 μm (a), 1 μm (b), 200 nm (c), 50 nm (d,e), 20 nm (f).

Figure R2 | The anelastic strain recovery in a single crystalline CuO NW. a,b, Low-magnified TEM images showing the shape recovery of a CuO NW after the bending stress released. Scale bar, 50 nm (a,b)

For cylindrical specimens with the diameter of d , the diffusivity D is related to the relaxation time τ by (Schaumann, G. *et al. Phys. Rev. Lett.* **21**, 891-893 (1968))

$$13.55D \tau = d^2$$

Take the NW shown in Fig. 1b for an example ($d \sim 46$ nm), it is evident that the non-linear Gorsky effect dominates during the anelastic strain recovery (Fig. R3), identical to that reported in ZnO NWs (Cheng, G. *et al. Nat. Nanotechnol.* **10**, 687-691 (2015)), suggesting that there are at least two kinds of point defects. Besides oxygen vacancy, Cu vacancy has been reported as another dominant type of point defect in CuO (Wu, D. *et al. Phys. Rev. B* **73**, 235206 (2006)). Hence, we have conducted piecewise linear fitting of the anelastic strain (with logarithm transform for y-coordinate) versus time (Fig. R3a) curves by both considering the migration of O and Cu vacancies. Since the slopes of linear fitting equal to $1/\tau_1$ and $1/\tau_2$, we obtained the average diffusivities of $D_1 = 4.09 \times 10^{-14}$ cm²/s (could be related with O vacancies) and $D_2 = 3.39 \times 10^{-15}$ cm²/s (could be related with Cu vacancies), close to the values reported for O vacancies and Zn interstitials (Cheng, G. *et al. Nat. Nanotechnol.* **10**, 687-691 (2015)). Because the experimental NWs are not central-symmetric, and the stress/strain is not evenly distributed along the NW length direction, the calculated values are not accurate. As suggested by the reviewer, we have incorporated the discussion of diffusivities in the revised manuscript (see also Fig. R3 (Supplementary Fig. 16)). (see line 1, paragraph 1, page 10 in the revised manuscript).

Figure R3 (Supplementary Figure 16) | Estimation of diffusivity of defects in a single crystalline CuO NW (diameter ~46 nm). a, Anelastic strain (with logarithm transform for

y-coordinate) as a function of time in six bending tests. Piecewise linear fitting conducted on all six bending tests. The slope equals to $1/\tau$. **b**, Diffusivities of two types of point defects. The orange and purple dashed lines indicate the averaged diffusivities of defects 1 and 2, respectively.

3. The nanowire in Figure 4 appears to be much smaller. Is it the same nanowire as the one in Figure 1?

Reply: They are not the same NW. The NW in Fig. 4 was fabricated by the top-down method, i.e. applying the focused e-beam to fabricate size-controllable NWs or nanobridges (*see page 13, Methods, Sample synthesis*). Due to the smaller lateral size, the CuO_x phase could firstly nucleate in the NW (Fig. 4) subjected to the nonuniaxial tensile stress (coupled tensile and bending stress) as a result of the thermal expansion induced by focused electron beam irradiation (Rodrigues, V. *et al. Phys. Rev. Lett.* **85**, 4124-4127 (2000); Zhao, P. *et al. Phys. Rev. Lett.* **123**, 216101 (2019)), as shown in Au and ZnO nanobridges (Fig. R4). Afterwards, the stress could be slowly released due to the lattice relaxation, resulting in the transformation from CuO_x phase to CuO. In this case, the shear strain evolution related with the phase transition can be more easily monitored.

Figure R4 | The existence of nonuniaxial tensile stress (coupled tensile and bending stress) in NWs fabricated by the top-down method. a,b, The relative displacement between left and right {111} planes (indicated by arrows) in Au nanobridge induced by nonuniaxial stress ((Rodrigues, V. *et al. Phys. Rev. Lett.* **85**, 4124-4127 (2000)). **c,d,** The anticlockwise rotation ($\sim 5^\circ$) of the atomic planes in a ZnO nanobridge caused by non-uniaxial tensile stress (Zhao, P. *et al. Phys. Rev. Lett.* **123**, 216101 (2019)).

4. The authors demonstrated that the new phase CuO_x was not stable. However, details were shown in Figures 3 and 4. It should be clarified that how the authors kept such phase in the nanowire for further study like EELS and EDS analysis. It should be mentioned (or more details should be given) if the experimental conditions were changed for different analysis.

Reply: Actually, the CuO_x phase could exist for tens of minutes, which depends on the e-beam intensity as well as the size of CuO_x phase. For example, applying the high-resolution TEM technique (e-beam intensity is $10\sim 10^2 \text{ A/cm}^2$), it takes more than

45 minutes when the CuO_x completely transformed back to CuO , leading to the complete strain recovery of the NW (Fig. 1e). Moreover, under the STEM image taking mode, the current density on the scan region is around $10\sim 10^2 \text{ A/cm}^2$, whereas the metastable phase could also exist for tens of minutes. As suggested by the reviewer, we have incorporated the information regarding the e-beam intensities for TEM and STEM modes (see *HRTEM and STEM/EELS experiments* in the Methods part) to facilitate a better understanding of the phase instability of CuO_x .

5. Surface-mediated migration of point defects could be dominated in nanowires during anelastic deformation. A remaining question that was not clearly addressed in this work is - does such migration of point defects promote the formation of the metastable phase CuO_x in the present study? For example, Figure 4 clearly verified the formation of the metastable phase CuO_x but it is unclear if the CuO_x is related to the anelastic deformation.

Reply: We agree that surface-mediated migration of point defects could also occur, which may lead to the nucleation of metastable CuO_x phase. Accordingly, SEM images show that the NW surfaces are faceted (Fig. R1). The TEM image indicate that the NW cross-sections are not circular and non-symmetric. Moreover, the NW surfaces are mainly consisted of several low-index lattice planes, e.g. (002) , (110) , $(1\bar{1}\bar{1})$, and $(\bar{1}\bar{1}\bar{1})$ etc (Fig. R1). Besides, the $(20\bar{2})$ plane was reported to be another possible surface plane (Yuan, L. *et al. Acta Mater.* **59**, 2491-2500 (2011)). Hence, we have calculated the migration energy barriers of oxygen vacancies within different surfaces. The relatively lower surface migration energy barriers (less than 1 eV, see Fig. R5) as compared with those calculated in the bulk (the lowest migration energy in Fig. 3 is $\sim 1.11 \text{ eV}$) suggests that the oxygen vacancies prefer to migrate along the surface area, which may result in the relatively larger anelasticity in the nano-sized material with higher aspect ratio. This is consistent with that reported in the reference (Cheng, G. *et al. Nat. Nanotechnol.* **10**, 687-691 (2015)). In addition, the stress-gradient driven vacancy migration within the NW should also occur because:

after stress releasing, it is consistently found that the CuO_x/CuO phase boundaries gradually move towards the free surface (Fig. R6 and Fig. 3). If the surface-mediated migration of point defects dominates or the surfaces serve as preferential sinks for vacancies, the CuO_x to CuO phase transition should initiate firstly at the free surface and the boundary should move from the free surface towards the inner side of the NW.

Figure R5 (Supplementary Figure 15) | CI-NEB calculated diffusion energy barriers of oxygen vacancies on CuO surface planes including (002) , (110) , $(11\bar{1})$, $(\bar{1}11)$, $(20\bar{2})$, (200) , (011) , and $(1\bar{1}2)$.

Figure R6 | The migration direction of boundaries between CuO_x and CuO phases. a-c, The phase boundaries between CuO_x and CuO gradually move towards the free surface. Scale bar, 2 nm (a-c).

The reversible phase transition from CuO to CuO_x was consistently found during the bending test, indicating that the nucleation and annihilation of CuO_x is closely

related with the anelastic strain. Further strain analysis of phase transition in Fig. 4 clearly shows the shear strain induced by the phase transformation.

As suggested by the reviewer, we have incorporated the discussion regarding the effects of NW surface/geometry on the vacancy migration (*see line 4, paragraph 1, page 9 in the revised manuscript*).

6. The authors gave an example showing the new phase formed very close to one end of the nanowire. For a bending test, the strain (or stress) distribution should be uniformly distributed along the nanowire length direction. Is there any evidence showing that the new phase is distributed along the nanowire length direction?

Reply: Ideally, for a uniformly bending NW, the new phase should uniformly distribute along the NW length direction. However, in the physical reality, it is extremely hard to realize the uniform strain distribution along the NW length due to the “non-ideal” shapes and structures of the as-fabricated NWs: (1) the NWs are not central-symmetric (Figs. R1-2); (2) the NW surface is not flat and consisted of surface steps; (3) the NW contains non-uniformly distributed defect structures, such as vacancies and dislocations, etc. Hence, under mechanically bending, the strain is not evenly distributed based on the atomic-scale perspective, and buckle would frequently occur in NWs subjected to large bending strains (Wang, S. *et al. Adv. Sci.* **4**, 1600332 (2017)). The phase transition initiates at the area with largest bending strain, whereas the oxygen vacancy concentration is large enough to trigger the nucleation of CuO_x phase. Upon further bending, the stress would concentrate on the phase-transformed area, resulting in the continuous growth of the CuO_x phase. Hence, we did not find the evenly distributed CuO_x phase along the NW length direction.

7. For the identification of new phase CuO_x , there is no direct experimental evidence to support that x is 0.67. It is suggested to use “metastable phase” to describe the experimental findings, although the stoichiometry of the new phase could be Cu_3O_2 .

Reply: As suggested by the reviewer, we use “metastable phase” or “CuO_x phase” to describe the nucleated oxygen-deficient Cu-O structure and propose Cu₃O₂ as a possible candidate of the metastable phase in the revised manuscript.

8. Page 11, line 208, “oxygen diffusion pathways ... are revealed”, where more accurately “revealed” is suggested changing to “proposed”.

Reply: As suggested by the reviewer, we have changed “revealed” to “proposed” (*see line 7, paragraph 1, page 13 in the revised manuscript*).

Reviewer #2:

In my opinion this is an interesting and original paper, worthy of publication. It is generally well written with only a few minor changes needed to the style of the English used.

1. The identification of the new Cu_3O_2 phase appears to be reasonably convincing, but it is the manner by which it is formed and the reversal back to CuO , that I have some issues with. In particular, the authors state that the nucleation of the new phase may be due to the Gorsky effect in which oxygen vacancies diffuse from the tensile to the compressive region of the sample during mechanical loading. They also claim that this movement is reversed after the load is removed. However, they appear to be neglecting the geometry of the nano-wire, with many free surfaces in the vicinity of the transformed region. These surfaces can act as sources and sinks for vacancies; hence on removal of the load, in particular, oxygen vacancies could readily move to these nearby surfaces, even at room temperature. Hence, I believe that the paper needs to be modified to take greater account of the role of the sample geometry in any discussion of the mechanisms.

Reply: We appreciate the reviewer for pointing out that “this is an interesting and original paper, worthy of publication”.

As suggested by the reviewer, the effects of NW surface/geometry on the vacancy migration are further investigated. Firstly, we found that the NW cross-sections are non-symmetric (not circular) and surfaces are faceted (Fig. R7), which are consisted of several low-index lattice planes, e.g. (002) , (110) , $(1\bar{1}\bar{1})$, and $(\bar{1}\bar{1}\bar{1})$ etc (Fig. R7). Besides, the $(20\bar{2})$ plane was reported to be another possible surface plane (Yuan, L. *et al. Acta Mater.* **59**, 2491-2500 (2011)). Hence, we have calculated the migration energy barriers of oxygen vacancies along different surfaces. The relatively lower calculated diffusion energy barriers of oxygen vacancies on different surface planes (e.g. (002) , (110) , $(1\bar{1}\bar{1})$, $(\bar{1}\bar{1}\bar{1})$, $(20\bar{2})$, (200) , (011) , and $(\bar{1}\bar{1}\bar{2})$) of CuO NW (less than 1 eV, Fig. R8 (Supplementary Fig. 15)) than those in the bulk (the

lowest migration energy in Fig. 3 is ~ 1.11 eV) suggests that the oxygen vacancies prefer to migrate along the surface area, which may result in the relatively larger anelasticity in the nano-sized material with higher aspect ratio. It has been reported that surface-mediated migration of point defects promotes Gorsky relaxation in ZnO NWs as well (Cheng, G. *et al. Nat. Nanotechnol.* **10**, 687-691 (2015)).

Figure R7 (Supplementary Figure 14) | The morphologies of CuO NWs. a-c SEM images showing the morphologies of fabricated CuO NWs. d-f, TEM images showing the cross-sections of the CuO NWs fabricated by focused ion beam (FIB). Scale bar, 2 μm (a), 1 μm (b), 200 nm (c), 50 nm (d,e), 20 nm (f).

Figure R8 (Supplementary Figure 15) | CI-NEB calculated diffusion energy barriers of oxygen vacancies on CuO surface planes including (002), (110), (111), (111), (202), (200), (011), and (112).

Besides, the stress-gradient driven vacancy migration within the NW should also occur because: after stress releasing, it is consistently found that the CuO_x/CuO phase boundaries gradually move towards the free surface (Fig. R9 and Fig. 3). If the surface-mediated migration of point defects dominates or the surfaces serve as preferential sinks for vacancies, the CuO_x to CuO phase transition should initiate firstly at the free surface and the boundary may move from the free surface towards the inner side of the NW, which was not observed in our experiments.

Figure R9 | The migration direction of boundaries between CuO_x and CuO phases. a-c, The phase boundaries between CuO_x and CuO gradually move towards the free surface. Scale bar, 2 nm (a-c).

As suggested by the reviewer, we have incorporated the discussion regarding the

effects of NW surface/geometry on the vacancy migration (*see line 4, paragraph 1, page 9 in the revised manuscript*).

2. Some comparison of this newly found Cu_3O_2 phase with the phases formed during the deposition of PLD films of copper oxide is also made in the paper. (Supplementary Fig. 10 and reference 1). I find this comparison unconvincing, since only one XRD peak shown in Fig. 10 is common in the two structures. Other peaks are found in one and not the other and vice versa. This could be explained, in part, by a strong texture in the deposited film, but although there may be some commonality between the two observations, I believe that the comparison is overstated in the paper.

Reply: We agree with the reviewer that the comparison between the simulated XRD pattern of Cu_3O_2 phase with the experimental pattern of copper oxide films is overstated in the paper. As indicated by the reviewer, besides some commonality between the two structures, the discrepancy could be explained, in part, by a strong texture in the deposited film. To avoid the misleading it may cause, in Fig. R10 (Supplementary Fig. 10), we identified the peaks of Cu_3O_2 in XRD data by blue vertical lines (Farhad, S. *et al. Materialia* **3**, 230-238 (2018)). And we have replaced “The similarity between the simulated X-ray diffraction (XRD) pattern based on present Cu_3O_2 structure and those reported in the literatures (Supplementary Fig. 10) suggests that the current structural model could possibly be applied in interpreting the oxidation mechanism in Cu and Cu_2O .” by “The calculated X-ray diffraction (XRD) patterns based on current Cu_3O_2 structural model and the experimental XRD pattern of pulsed laser deposited cuprous oxide thin films³⁶ are displayed in Supplementary Fig. 10. We found that although there might be some similarity between the two structures, the peaks do not match quite well between the two phases, possibly due to the strong texture in the experimental deposited film. Further efforts are required to reveal the structural relationship between current metastable phase and Cu_3O_2 phase reported in the literatures³⁴⁻³⁶.” (*see line 3, paragraph 1, page 7 in the revised manuscript*)

Figure R10 (Supplementary Figure 10) | Experimental XRD pattern of the reported Cu₃O₂ and Cu₂O (blue line) (Cu₂O and Cu₃O₂ peaks are presented by black and blue vertical lines, respectively) in reference and the simulated XRD pattern based on the monoclinic Cu₃O₂ structure (red vertical lines) in this paper.

REVIEWERS' COMMENTS

Reviewer #1 (Remarks to the Author):

The authors have satisfactorily addressed this reviewer's comments. I recommend publication of this manuscript. One minor, non-mandatory suggestion: The authors might consider removing "yet the descriptions of its origin remain phenomenological" in Introduction because as the authors wrote "It was speculated that ...". The origin is not phenomenological, rather physics-based. The limitation in that work was no direct atomistic observation. Similarly, I would suggest the authors cite reference 9, to avoid confusion, after "... contribute to a much larger anelastic strain, even at room temperature."

Response to the reviewer

Reviewer #1

The authors have satisfactorily addressed this reviewer's comments. I recommend publication of this manuscript. One minor, non-mandatory suggestion: The authors might consider removing "yet the descriptions of its origin remain phenomenological" in Introduction because as the authors wrote "It was speculated that ...". The origin is not phenomenological, rather physics-based. The limitation in that work was no direct atomistic observation. Similarly, I would suggest the authors cite reference 9, to avoid confusion, after "... contribute to a much larger anelastic strain, even at room temperature."

Reply: We have removed "yet the descriptions of its origin remain phenomenological" in Introduction and added the reference 9 in the mentioned position according to the reviewer's suggestion.